# The Effects of Soccer Specific Exercise on Countermovement Jump Performance in Elite Youth Soccer Players

**DOI:** 10.3390/children9121861

**Published:** 2022-11-30

**Authors:** Max Lyons Donegan, Steven Eustace, Rhys Morris, Ryan Penny, Jason Tallis

**Affiliations:** Centre for Sport Exercise & Life Sciences, Coventry University, Coventry CV1 5FB, UK

**Keywords:** countermovement jump, neuromuscular fatigue, statistical parametric mapping, youth soccer

## Abstract

The aims of the study were to examine the test–retest reliability of force-time (F-T) characteristics and F-T curve waveform of bilateral and unilateral countermovement jumps (CMJ) in elite youth soccer players and to evaluate the effects of competitive match-play on CMJ performance. 16 male youth soccer players completed CMJs on two separate occasions to determine reliability, and immediately pre, post and 48 h following a competitive match. Coefficient of variation (CV%), Intra-class correlation coefficient (ICC) and limits of agreement were used to assess reliability of discreate CMJ variables. Single factor repeated measures ANOVA were used to determine the effects of match play. Statistical parametric mapping was used to evaluate the repeatability of the CMJ force-time waveform and the effects of match play. Jump height had limited reliability in all three jumps and only a select few jump specific F-T variables were found to be reliable (CV < 10%, ICC > 0.5). Select variables were reduced immediately post game but recovered 48 h post game. The F-T curve waveform was found to be repeatable but did not differ following match-play. This study suggest that select F-T variables change following match-play and may be suitable tools to allow practitioners to detect decrements in performance. These data may help inform practitioners to use the most appropriate F-T variables to assess fatigue and recovery, with implications for performance and injury risk.

## 1. Introduction

The use of the countermovement jump (CMJ) is common practice in professional soccer [1] due to its ease of implementation, the richness of information and low risk of injury [2,3]. The increased affordability and popularity of force platforms have become essential in promoting the diverse application of CMJ assessment within the literature. The CMJ has been used as a lower limb exercise [4], a benchmarking tool for the assessment of lower limb power [5] and evaluation of neuromuscular function [6,7] and fatigue responses [2]. Whilst the detection of neuromuscular fatigue is reliable [2]; this evidence is derived from adult professional athletes, and transferability to youth professional soccer players is yet to be established.

CMJ performance depends on the neuromuscular interaction of the entire kinetic chain during the sequence of events leading to take-off [1]. Effective jump performance requires precise sequencing of eccentric and concentric muscle actions to maximise the stretch-shortening cycle (SSC) contribution to generating vertical propulsion [8]. Force platforms provide an opportunity to understand the kinematic process underpinning jump height, giving insight into the neuromuscular function of an athlete. Understanding an athlete’s neuromuscular function is essential, particularly in youth athletes [9], to ensure practitioners provide the correct training stimulus to allow the athletes to tolerate the demands of both long-term training and competitive match play [10].

Given the breadth of data that can be determined from CMJ F-T analysis, there are challenges in selecting meaningful outcome variables and the reliability of such variables among different populations. The test–retest reliability of CMJ F-T derived outcome variables is needed to observe differences more confidently between different athletes’ measures and monitor performance over time. The test–retest reliability of CMJ height has been well established in elite adult sporting populations [11], and whilst there is little consensus on which F-T characteristics should be monitored and reported [12,13]. Vertical jump capabilities are fundamental to motor skill requirements in soccer and while bilateral CMJs are commonly used assessments among both adult and youth athletes, soccer performance requires unilateral and bilateral vertical propulsion, and the majority of maximal actions in soccer are unilateral, highlighting the importance of assessing vertical jump performance [14]; assessing however, there is a distinct lack of evidence exploring the reliability of unilateral CMJ F-T characteristics in youth populations.

CMJ assessments have become an important measure to assess the impact of training load given their proposed sensitivity to detect neuromuscular fatigue among adults [2]. Neuromuscular fatigue can be defined as any exercise-induced reduction in maximal voluntary force [15], in the case of this research, it is defined as a change in outcome variables and/or changes in strategy to elicit jump performance. However, there is little consensus regarding the most appropriate F-T variables to measure to detect neuromuscular fatigue during CMJ assessments [10,16]. While conflicting findings have been reported in regard to whether CMJ performance is affected by fatigue or sensitive enough to detect fatigue [1,12,13], there are differences between the evidence.

Solely observing discrete variables, such as jump height, may be limited given that they are derived from a single data point on a F-T curve and fails to account for the F-T curve in its entirety [2]. This means that practitioners may be left unaware of changes in strategy due to fatigue, despite the maintenance of outcome variables [17]. Examining changes in the shape of the F-T curve may accurately portray a change in jump strategy that might be associated with neuromuscular fatigue or not [2]. One approach to this could be the application of statistical parametric mapping (SPM) to the F-T data obtained during the contact phase. SPM is a method of analysing 1D continuous statistical data by interpreting peaks or clusters as regionally specific effects attributable to the experimental manipulation [18].

Given the identified gaps in the literature, the present study aimed to examine the reliability of F-T characteristics of bilateral and unilateral CMJs in elite youth soccer players. By utilising the calculation of discrete F-T measures and SPM, the project further aimed to evaluate the effects of competitive match play on bilateral and unilateral CMJ performance in this population.

## 2. Materials and Methods

16 male youth soccer players (age: 15.2 ± 2.4 years, height: 171.3 ± 21.3 cm, body mass: 59.4 ± 24.6 kg) agreed to participate in this study. All participants were part of the same category two premier league academy and trained three times per week, playing in at least one competitive match weekly. A total of 16 participants completed the reliability aim of the study (Study A) and 12 completed the producers used to evaluate the effect of exercise (Study B).

### 2.1. Study A

Participants attended two separate testing sessions at the same time in order to account for circadian rhythms, separated by five days [19]. Following a standardised dynamic warm-up consisting of squats, lunges and CMJ’s, participants performed three trials [19] of CMJ_BI_, CMJ_DOM_ and CMJ_ND_. Each trial was separated by two minutes of passive recovery [20]. Before executing each CMJ, and participants were instructed to “jump as high and fast as possible” with hands fixed on their hips [7]. A total of 14 F-T outcome variables were observed; Jump Height(m), modified Reactive Strength Index (mRSI), Force at Minimum Displacement (Newtons), Countermovement Depth (m), Time to Takeoff (s), Eccentric Time (s), Peak Eccentric Power (W), Mean Eccentric Power (W), Eccentric Impulse (kg·m/s), Concentric Time (s), Peak Concentric Power (W), Mean Concentric Power (W), Concentric Impulse (kg·m/s) and Eccentric:Concentric Ratio. The above outcome variables were chosen to give practitioners and researchers a detailed understanding of the output, drivers, and strategy of CMJ performance [21], whilst also giving equal consideration to the concentric and eccentric portions of the movement relationship between the two. These markers have also been used in previous studies as performance markers and in the assessment of neuromuscular fatigue [11,22].

### 2.2. Study B

Participants attended three separate testing sessions and completed three trials of CMJ_BI_, CMJ_DOM_ and CMJ_ND_. The testing sessions were pre game, immediately post-game (IPG) and 48 h post game (48Post) (Figure 1). The fatiguing protocol was the competitive match play against opposing Premier League Category 2 academies. The opposing clubs faced were of the same standard within the Elite Player Performance Plan structure. The length and rules of the match varied depending on the age group, but all matches followed Premier League guidelines. Games for participants in the U16 age group (n = 5) consisted of two halves of 80 min, games for players in the U14 age group (n = 7) consisted of two halves of 35 min, all participants played the entire game. 

### 2.3. Data Analysis

F-T data was collected using portable force plates (Wireless Dual Force Platform V.3.0, Hawkin Dynamics, Westbrook, ME, USA) using a sampling rate of 1000 Hz. A web application and manufacturers guidelines (Wireless Dual Force Platform V.3.0, Hawkin Dynamics, ME, USA) was used to view results and export raw force time data and discrete variables calculated by the software. Concentric and eccentric phases of the movement were differentiated using the manufacturers guidelines (Wireless Dual Force Platform V.3.0, Hawkin Dynamics, ME, USA). The propulsive phase of the F-T curve waveform was taken, starting force (N) was 5 times higher than the standard deviation in body mass, and ending force was less than 10 N [23]. The curve was then normalized to stance using customized script in MATLAB (R2019a, version 9.6.0.1072779, The MathWorks, Inc., Natick, MA, USA).

### 2.4. Statistical Analysis

The discrete F-T variables were averaged over three successful trials for each participant to indicate performance for each CMJ assessment [24]. To determine between-session reliability, paired *t*-test intraclass correlation coefficient (ICC) was calculated using IBM SPSS Statistics (IBM, 2019), coefficient variation (CV). Graphical representation of 95% Bland–Altman plots was produced in R [25] using the ggplot2 package (v3.3.6; [26]), where heteroscedastic data were log-transformed (Appendix A). Paired *t*-tests were used to determine systematic bias between the different time points, interclass correlation coefficient (ICC’s) were used to ascertain the rater bias between these two-time points, and Bland–Altman limits of agreement (LoA’s) were used to determine the agreement between these two time points [26]. The ICC’s performed in this present study are interpreted as 0.2 = slight, 0.21–0.4 = fair, 0.41–0.6 = moderate, 0.61–0.8 = substantial, and 0.8 = almost perfect reliability [27]. The CVs were interpreted as reliable if found to be less than 10% [2]. To evaluate the effects of soccer match play, a single factor repeated measures ANOVA was conducted, partial eta squared (ηp2), and Cohen’s d (d) was calculated to determine effect size. Significant differences and main effects were explored using Bonferroni corrected pairwise comparisons, and partial eta squared was also calculated to determine effect size (0.01 = small, 0.06 = medium, 0.14 = high) [28].

The reliability of F-T curve was assessed using statistical parametric mapping (SPM), after trials were normalized to time, and *t*-tests were performed to determine the alpha level (*p* value) between the two-time conditions. The reliability of F-T curve shape was further assessed by calculating individual ICCs using time-normalized data in IMB SPSS Statistics (IBM, 2019), which were reported as a range. ICCs were then reported as a group range, with individual ICC’s illustrated by a histogram (Figure 2). To identify the effect of time on kinetic waveforms (normalized to time) during the unweighting, braking and propulsive phases (Hawkin Dynamics, 2021), SPM one-way ANOVAs were performed in MATLAB (R2019a, version 9.6.0.1072779, The MathWorks, Inc., Natick, MA, USA) and implemented using the open-source one-dimensional SPM code (spm1D-package, version 0.4.3, http://spm1d.org/index.html (accessed on 12th January 2022)). If SPM{t} crossed the critical threshold, a supra- or infra-threshold cluster depicted by grey shading indicated a significant difference between given time points during the take-off phase of a CMJ. Data is reported as mean and standard deviation, with statistical significance set at *p* ≤ 0.05.

## 3. Results

As identified by Table 1, the reliability of discrete F-T characteristics for CMJ_BI_ varied with CVs ranging from 8–19%, ICCs ranged from 0.118–0.818. Peak Concentric (Con) Power and ConImpulse were found to be reliable (CV < 10%, ICC > 0.6) and showed varying levels of agreement in the Bland–Altman tests. Jump height exhibited limited reliability (CV = 13%, ICC = 0.488). The reliability for CMJ_DOM_ varied with CVs ranging from 9–25%, ICCs ranged from 0.176–0.854. Force at Minimum (Min) Displacement was found to be reliable (CV < 10%, ICC > 0.6). During CMJ_DOM_ performance and showed varying levels of agreement in Bland–Altman tests, jump height demonstrated limited reliability (CV = 18%, ICC = 0.527). For CMJ_ND_, the reliability varied with CVs ranging from 8–42%, ICCs ranged from 0.335–0.854. Jump height had limited reliability (CV = 21%, ICC = 0.357). Time to take off, ConTime, Peak Con Power and ConImpulse were reliable (CV < 10%, ICC > 0.6) during CMJ_ND_ performance and showed varying levels of agreement in Bland–Altman tests.

As identified by Figure 2, 14 SPM showed no significant differences in CMJ_BI_ F-T curve shape and moderate reliability between sessions (*p* > 0.05); ICCs > 0.616. Similarly, CMJ_DOM_ performance revealed no significant differences and moderate reliability for any participants (*p* < 0.05); ICCs > 0.5. There were also no significant differences in CMJ_ND_ F-T curve shape and moderate reliability (*p* < 0.05); ICCs > 0.5.

For discrete measures of EccImpulse and ConImpulse determined from CMJ_BI_, there was a significant main effect of time with large effect size (Table 2.; *p* ≤ 0.03. np2 = 0.276, 0.369). Post hoc analysis revealed that EccImpulse and ConImpulse decreased with a large effect size between measures pre and IPG (*p* < 0.05; d = 0.09, 0.04) and then increased back to baseline between measures IPG and 48Post (*p* < 0.05; d = 0.04, 0.03). There were no other statistically significant differences Similarly, force at Min Displacement and Mean Ecc Power determined from CMJ_DOM_ demonstrated a significant main effect of time with a large effect size (Table 2. *p* < 0.04. np2 = 0.335, 0.271). Post hoc analysis revealed that force at Min Displacement decreased significantly with a large effect size between measures pre and IPG (*p* < 0.05; d = 0.09), and then increased significantly back to baseline between measures IPG and 48Post (*p* < 0.05; d = 0.05). Mean Ecc Power increased significantly between measures IPG and 48Post (*p* < 0.05; d = 0.01). There were no other statistically significant differences. However, for CMJ_ND_ performance, there were no statistically significant differences (*p* > 0.05).

One-way repeated measures ANOVA SPM indicated inter-threshold clusters in CMJ_BI_, CMJ_DOM_ and CMJ_ND_ (Figure 3). No cluster exceeded the threshold in any analysis’s main effect of time. 

## 4. Discussion

The present study aimed to determine the reliability of bilateral and unilateral CMJ assessments in elite youth soccer players and evaluate the efficacy of CMJ assessments in detecting exercise-induced neuromuscular fatigue due to soccer match play. Jump height had limited reliability across assessments; however, specific discrete force-time (F-T) metrics (Peak Con Power and ConImpulse for CMJ_BI_, Force at Min. Displacement for CMJ_DOM_, and Time to take off, ConTime, Peak Con Power and ConImpulse for CMJ_ND_) were reliable (CV < 10, ICC > 0.5), although these were specific to CMJ_BI_, CMJ_DOM_ or CMJ_ND_ performance. Immediately following and 48 h post soccer match play, CMJ jump height was unchanged. However, other discrete F-T characteristics (Eccentric and Concentric Impulse for CMJ_BI_ and Force at Min. Displacement and Mean Ecc Power for CMJ_DOM_) were influenced by soccer match-play. These were also specific to CMJ_BI_, CMJ_DOM_ or CMJ_ND_ performance. All but Mean Ecc Power decreased between measures pre and IPG and then increased in return to baseline measures between measures IPG and 48Post, Mean Ecc Power increased significantly between measures IPG and 48Post. SPM indicated that the F-T curve over the propulsive phase was unaffected post-soccer match play. These findings suggest that caution is needed when using CMJ as an assessment tool in youth athletes and that only a select few F-T variables used in this study may effectively detect NMF in the 48 h following match play.

The lack of repeatability in measures of jump height in CMJ_BI_, CMJ_DOM_ and CMJ_ND_ performance in the current study is contradictory to that of previous research studies in youth [22,28,29,30], likely due to lack of maturity related adaptations among the current participants [31]. Previous research has found that prepubertal athletes improve physical performance mainly through neuromuscular improvements [32] due to the plasticity of their central nervous system [33]. Plyometric training is often used as a maturity-dependent intervention to promote ‘synergistic adaptation’, a symbiotic relationship between interventions and maturity-related adaptations [31]; in this case, motor recruitment, contraction velocity, and pre-activation, which are all utilized in CMJ performance [34]. This would suggest that the participants in this study are yet to undergo these adaptations to the neuromuscular system, which will result in better quality mechanics of the SSC and more repeatable jump performance [34].

Whilst the variables that were found to be reliable, did vary between jump types, and jump height was found to lack repeatability in all three jump types. It was found to be most reliable in CMJ_BI_ and least reliable during CMJ_ND_ performance. During CMJ_BI_ performance, the load placed on each limb is reduced, and the power or force produced by each limb is less than when each limb is used in unilateral CMJ performance; this is termed bilateral deficit [35] and explains the more repeatable performance outcome in bilateral CMJ. The opposite is then true for unilateral CMJ performance, where a greater load is placed upon an individual limb, and greater force is required when using limbs individually, often resulting in less controlled and more volatile outcomes [35]; this is exacerbated with youth athletes, particularly those at early stages of biological development.

Using a combination of SPM and ICCs, data from the present study indicates, for the first time, that CMJ_BI_, CMJ_DOM_ and CMJ_ND_ F-T curve waveform is reliable in a non-adult population, something that has previously been observed in CMI_BI_ in adult populations [24]. The study’s findings suggest that CMJ_BI_, CMJ_DOM_, and CMJ_ND_ assessments must be considered separately and used with caution, given their poor reliability among youth athletes. The current study also aimed to evaluate the efficacy of CMJ assessments in detecting exercise-induced neuromuscular fatigue due to soccer match play. The findings during CMJ_BI_ performance align with previous research among adult populations, which found little effect of time on CMJ jump height following both simulated and competitive soccer match play [1,36,37]. However, the current literature is ambiguous as many studies have found contradictory findings [12,13], such as decreased jump height 24 h after a fatiguing protocol and an increase in a return to baseline measures 48 post protocol [12]. The current study is thought to be among the first to observe this among an elite youth soccer population and found changes in jump-specific F-T variables, which suggests a change in CMJ strategy to maintain performance and jump height [1]. Something that has been observed previously in high-level performers [38]. This research is among the first to assess the effects of an exercise protocol on unilateral CMJ performance in this population.

Data from the present study indicated that the F-T curve waveform following the exercise protocol was unchanged, which contradicts previous research using CMJ_BI_ in an adult population [24]. It has been suggested that CMJ movement patterns are altered due to fatigue [17] and result in variations of F-T curves (where curves have been normalized to time and compared at each time point), even when there is no significant difference observed in jump height [7]. The dynamical systems theory explains the ability to alter muscle recruitment, where multiple joints and muscles come up with different movement solutions to complete desired movement outcomes [24], and the ability to do this may be dependent on skill level [39,40]. In the case of this research, it may be that the youth athletes are yet to reach a skill level where they can alter muscle recruitment to maintain performance, the lack of repeatability among performance outcome measures in this study further demonstrates a lack of required skill among participants. These findings may further be explained by the primary strength phases proposed by Suchomel, Nimphius and Stone (2016). The first phase of strength, the strength deficit phase, is based on the motor learning of the individual and is described as a phase where individuals are improving their ability to generate force but are not yet able to exploit their strength and translate it into improved performance [41]. Athletes within this phase often develop physical literacy, which is understood to be the case in pre-pubertal athletes and athletes going through growth-related changes [32,41]. Given the current population anlaysed in this research were youth athletes, it is plausible the above theories explain the lack of repeatable findings for CMJ performance. 

## 5. Conclusions

In conclusion, the findings of this study suggest that select jump-specific F-T outcome variables change following soccer-specific exercise-induced and may be suitable tools to allow practitioners to detect decrements in performance among their athletes. However, when examining exercise induced-fatigue, practitioners should appreciate that most F-T variables show low reproducibility, specifically jump height, which is most commonly used to monitor performance. Therefore, a large bandwidth for change needs to be considered when assessing changes. There are also markedly different effects between jump types, with unilateral CMJ showing lower reproducibility and different F-T variables that can detect exercise-induced changes. Finally, exercise-induced fatigue does not appear to manifest in changes to F-T curve waveform and so SPM does not appear to offer any insight into changes in CMJ strategy that may not have been apparent through observing performance in elite youth soccer players. 

## Figures and Tables

**Figure 1 children-09-01861-f001:**
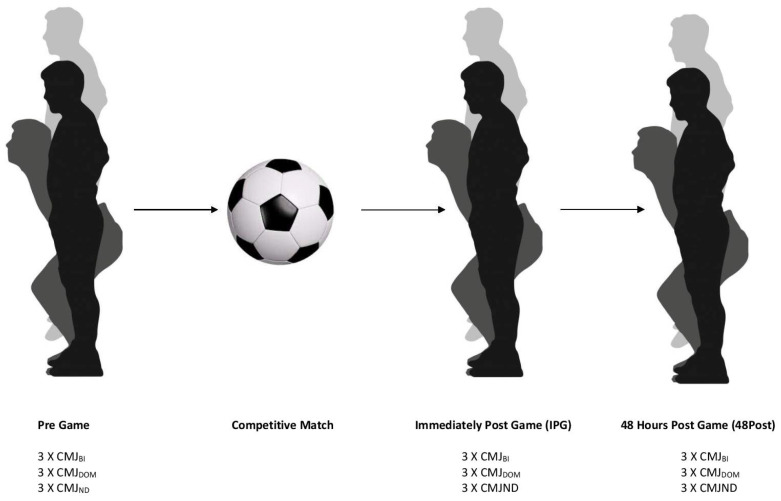
Visualization of experimental design of Study B.

**Figure 2 children-09-01861-f002:**
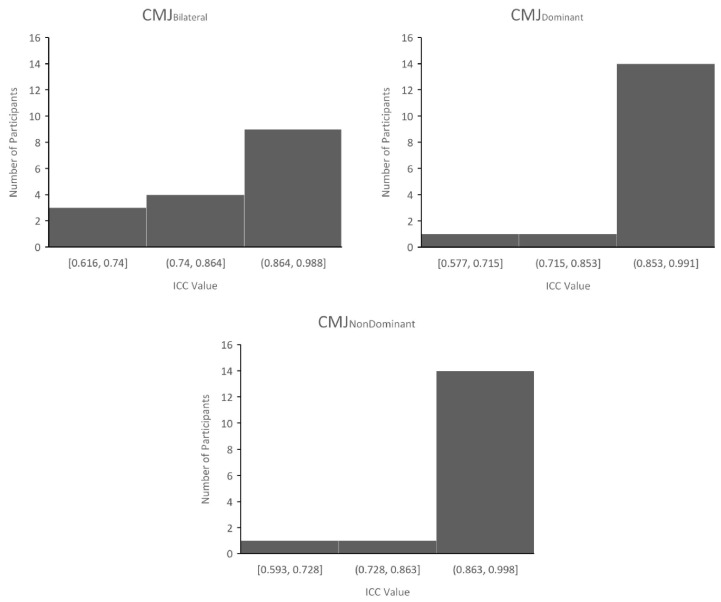
Range and spread of individual ICC values of force-time curve shape in CMJBI, CMJDOM and CMJND.

**Figure 3 children-09-01861-f003:**
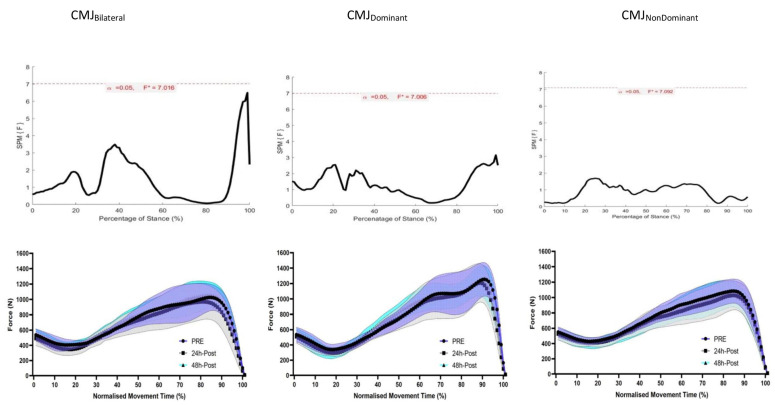
SPM 1-way repeated measures ANOVA results for CMJ_BI_, CMJ_DOM_ and CMJ_ND_ (top panels) where the SPM {t*} curve exceeds the critical threshold (dotted line), this area is shaded, and a statistically significant relationship is present at those nodes. Mean and SD force-time curve waveforms (bottom panels) for performance Pre, Post and IPG during CMJ_BI_, CMJ_DOM_ and CMJ_ND_ performance.

**Table 1 children-09-01861-t001:** Test Re-Test reliability of CMJ_BI_, CMJ_DOM_ and CMJ_ND._

						Bland–Altman		
	Outcome	Test 1	Test 2	CV (%)	ICC (95% CI)	Bias	Lower LOA	Upper LOA
CMJBI	Jump Height (m)	0.30 ± 0.05	0.29 ± 0.05	13.16	0.49 (0.07 to 0.88)	0.01 ± 0.07	−0.12	0.13
	mRSI	0.29 ± 0.06	0.30 ± 0.08	15.78	0.54 (0.31 to 0.92)	−0.01 ± 0.09	−0.17	0.16
	Force at Min. Displacement (N)	1133.46 ± 200.47	1064.35 ± 181.47	13.52	0.68 (0.67 to 0.960	−69.10 ± 249.64	−562.86	424.65
	Countermovement Depth (m)	0.33 ± 0.06	0.32 ± 0.06	11.37	0.70 (0.54 to 0.87)	−0.01 ± 0.05	−0.12	0.11
	Time to Takeoff (s)	1.08 ± 0.21	1.05 ± 0.14	8.89	0.59 (0.52 to 0.94)	0.02 ± 0.17	−0.35	0.30
	Ecc Time (s)	0.30 ± 0.12	0.28 ± 0.07	19.24	0.66 (0.04 to 0.89)	0.02 ± 0.11	−0.20	0.23
	Peak Ecc Power (W)	738.25 ± 291.74	660.24 ± 201.77	18.90	0.75 (0.60 to 0.95)	−78.01 ± 221.79	−526.44	370.42
	Mean Ecc Power (W)	541.40 ± 192.60	484.59 ± 142.42	19.52	0.75 (0.65 to 0.96)	−56.81 ± 149.32	−347.32	260.68
	Ecc Impulse (kg⋅m/s)	233.40 ± 59.59	215.46 ± 56.03	13.84	0.82 (0.45 to 0.94)	17.93 ± 53.60	−97.61	133.48
	Con Time (s)	0.34 ± 0.08	0.33 ± 0.06	12.37	0.63 (0.45 to 0.98)	0.01 ± 0.08	−0.13	0.16
	Peak Con Power (W)	2936.52 ± 492.26	2847.97 ± 443.69	8.62	0.67 (0.46 to 0.98)	88.54 ± 420.62	−722.77	899.87
	Mean Con Power (W)	1401.67 ± 249.65	1353.64 ± 274.98	12.06	0.53 (0.37 to 0.97)	−48.03 ± 306.70	−668.64	572.57
	Con Impulse (kg⋅m/s)	356.90 ± 65.50	335.33 ± 59.68	8.14	0.66 (0.48 to 0.96)	21.56 ± 48.30	−76.60	119.74
	Ecc:Con Ratio	1 ± 0.21	1 ± 0.14	14.20	0.12 (0.06 to 0.73)	0.01 ± 0.24	−0.47	0.45
CMJDOM	Jump Height (m)	0.14 ± 0.04	0.13 ± 0.03	18.40	0.53 (0.24 to 0.85)	0.00 ± 0.06	−0.10	0.10
	mRSI	0.15 ± 0.04	0.16 ± 0.06	15.79	0.70 (0.54 to 0.95)	−22.95 ± 41.73	−104.63	58.73
	Force at Min. Displacement (N)	940.33 ± 157.13	958.53 ± 183.71	8.99	0.80 (0.76 to 0.98)	−18.19 ± 140.25	−284.83	248.43
	Countermovement Depth (m)	0.18 ± 0.06	0.20 ± 0.04	13.45	0.76 (0.30 to 0.91)	0.01 ± 0.04	−0.07	0.10
	Time to Takeoff (s)	0.94 ± 0.21	1.02 ± 0.19	13.84	0.60 (0.45 to 0.86)	0.08 ± 0.23	−0.39	0.55
	Ecc Time (s)	0.23 ± 0.10	0.30 ± 0.09	25.15	0.36 (0.23 to 0.80)	−0.06 ± 0.16	−0.37	0.24
	Peak Ecc Power (W)	432.33 ± 150.86	437.19 ± 102.71	11.93	0.85 (0.78 to 0.96)	4.86 ± 94.86	−181.05	190.77
	Mean Ecc Power (W)	303.50 ± 117.23	307.89 ± 78.72	20.96	0.77 (0.69 to 0.96)	4.37 ± 89.17	−167.49	176.25
	Ecc Impulse (kg⋅m/s)	176.03 ± 59.13	219.32 ± 66.34	21.12	0.33 (0.17 to 0.65)	−14.36 ± 145.62	−192.62	163.89
	Con Time (s)	0.32 ± 0.09	0.36 ± 0.09	12.67	0.71 (0.16 to 0.90)	−0.03 ± 0.08	−0.21	0.14
	Peak Con Power (W)	1752.27 ± 415.52	1781.39 ± 335.05	11.11	0.77 (0.63 to 0.98)	29.12 ± 324.92	−606.89	665.14
	Mean Con Power (W)	885.77 ± 184.09	880.73 ± 220.75	12.89	0.74 (0.68 to 0.96)	5.03 ± 170.87	−329.52	329.52
	Con Impulse (kg⋅m/s)	293.96 ± 67.93	312.50 ± 56.57	10.70	0.79 (0.59 to 0.92)	−2.86 ± 104.21	−47.22	41.49
	Ecc:Con Ratio	1 ± 0.18	1 ± 0.18	18.55	0.18 (0.06 to 0.71)	−0.11 ± 0.44	−0.95	0.72
CMJND	Jump Height (m)	0.15 ± 0.02	0.16 ± 0.06	21.35	0.36 (0.16 to 0.85)	−0.01 ± 0.06	−0.11	0.09
	mRSI	0.17 ± 0.04	0.19 ± 0.08	24.06	0.40 (0.26 to 0.86)	0.01 ± 0.09	−0.16	0.13
	Force at Min. Displacement (N)	969.52 ± 117.94	885.39 ± 131.54	10.93	0.34 (0.21 to 0.96)	84.14 ± 156.70	−223.45	391.72
	Countermovement Depth (m)	0.18 ± 0.05	0.20 ± 0.08	19.51	0.63 (0.36 to 0.70)	0.02 ± 0.08	−0.16	0.19
	Time to Takeoff (s)	0.91 ± 0.21	0.96 ± 0.21	9.87	0.76 (0.51 to 0.94)	−0.04 ± 0.17	−0.41	0.31
	Ecc Time (s)	0.22 ± 0.10	0.25 ± 0.10	16.13	0.70 (0.42 to 0.93)	−0.02 ± 0.09	−0.21	0.16
	Peak Ecc Power (W)	447.04 ± 130.76	469.99 ± 285.98	21.79	0.45 (0.18 to 0.66)	−18.93 ± 283.40	−581.43	619.30
	Mean Ecc Power (W)	319.49 ± 92.94	300.14 ± 106.47	20.44	0.62 (0.22 to 0.86)	−16.38 ± 116.47	−207.75	174.98
	Ecc Impulse (kg⋅m/s)	172.26 ± 65.64	165.63 ± 55.53	13.12	0.75 (0.28 to 0.92)	−3.77 ± 93.75	−106.55	98.99
	Con Time (s)	0.31 ± 0.07	0.33 ± 0.08	8.96	0.84 (0.54 to 0.94)	−0.02 ± 0.06	−0.14	0.09
	Peak Con Power (W)	1884.79 ± 271.91	1860.48 ± 435.77	7.63	0.86 (0.72 to 0.97)	27.70 ± 391.97	−468.16	523.56
	Mean Con Power (W)	958.19 ± 129.65	904.91 ± 226.59	11.57	0.72 (0.65 to 0.97)	49.32 ± 225.54	−288.40	387.04
	Con Impulse (kg⋅m/s)	296.24 ± 61.49	284.65 ± 62.80	9.35	0.85 (0.57 to 0.82)	6.12 ± 89.73	−73.89	86.15
	Ecc:Con Ratio	1 ± 0.15	1 ± 0.15	10.69	0.74 (0.26 to 0.91)	−0.03 ± 0.15	−0.32	0.25

Data Represented as Mean ± SD. Abbrevaitions: mRSI = modified reactive strength index. Ecc = Eccentric. Con = Concentric. CV = Coefficient of Variation. ICC = Intraclass Correlation Coefficient. LOA = Limit of Agreement. m = metres. N = Newtons. s = seconds. W = Watt. kg⋅m/s = kilogram meter per second. CI= confidence interval.

**Table 2 children-09-01861-t002:** The effects of neuromuscular fatigue on CMJ_BI_, CMJ_DOM_ and CMJ_ND_.

		Mean				
	Outcome	Pre	IPG	48Post	*p*	np2
CMJBI	Jump Height (m)	0.28 ± 0.06	0.26 ± 0.07	0.26 ± 0.06	0.48	0.06
	mRSI	0.29 ± 0.10	0.28 ± 0.09	0.27 ± 0.08	0.87	0.02
	Force at Min. Displacement (N)	1077.81 ± 360.89	1003.82 ± 397.44	1067.50 ± 262.22	0.30	0.10
	Countermovement Depth (m)	0.33 ± 0.07	0.34 ± 0.10	0.35 ± 0.07	0.83	0.01
	Time to Takeoff (s)	1.02 ± 0.16	0.97 ± 0.19	0.97 ± 0.10	0.32	0.10
	Ecc Time (s)	0.27 ± 0.10	0.26 ± 0.08	0.28 ± 0.08	0.68	0.03
	Peak Ecc Power (W)	683.99 ± 280.41	655.56 ± 293.10	757.82 ± 290.95	0.15	0.16
	Mean Ecc Power (W)	505.37 ± 212.54	480.29 ± 222.36	557.62 ± 215.67	0.13	0.18
	Ecc Impulse (kg⋅m/s)	212.10 ± 63.57 *	192.40 ± 53.22^	226.74 ± 61.05	**0.03**	0.28
	Con Time (s)	0.32 ± 0.07	0.31 ± 0.05	0.31 ± 0.03	0.52	0.05
	Peak Con Power (W)	2651.86 ± 815.71	2551.26 ± 1103.45	2582.53 ± 814.17	0.59	0.04
	Mean Con Power (W)	1297.40 ± 502.62	1204.93 ± 585.76	1247.33 ± 394.02	0.48	0.07
	Con Impulse (kg⋅m/s)	318.75 ± 59.14 *	292.66 ± 82.05^	317.66 ± 65.61	**0.01**	0.37
	Ecc:Con Ratio	1 ± 0.27	1 ± 0.18	1 ± 0.21	0.43	0.07
CMJDOM	Jump Height (m)	0.13 ± 0.04	0.14 ± 0.05	0.12 ± 0.05	0.37	0.08
	mRSI	0.15 ± 0.08	0.17 ± 0.09	0.14 ± 0.08	0.77	0.09
	Force at Min. Displacement (N)	896.35 ± 328.18 *	823.67 ± 337.45 ^	945.25 ± 242.31	**0.02**	0.34
	Countermovement Depth (m)	0.20 ± 0.06	0.19 ± 0.10	0.19 ± 0.03	0.35	0.02
	Time to Takeoff (s)	0.97 ± 0.22	0.95 ± 0.26	0.93 ± 0.17	0.66	0.03
	Ecc Time (s)	0.26 ± 0.11	0.26 ± 0.14	0.22 ± 0.06	0.31	0.10
	Peak Ecc Power (W)	430.61 ± 149.13	388.56 ± 186.34	453.36 ± 146.59	0.10	0.19
	Mean Ecc Power (W)	303.51 ± 120.38	267.22 ± 138.71 ^	325.93 ± 106.50	**0.03**	0.27
	Ecc Impulse (kg⋅m/s)	181.74 ± 58.87	167.64 ± 69.42	170.31 ± 41.04	0.53	0.06
	Con Time (s)	0.37 ± 0.09	0.36 ± 69.42	0.34 ± 0.10	0.40	0.08
	Peak Con Power (W)	1581.17 ± 614.00	1583.98 ± 711.96	1586.23 ± 692.09	1.00	0.00
	Mean Con Power (W)	798.18 ± 373.50	772.83 ± 406.08	816.85 ± 378.58	0.40	0.08
	Con Impulse (kg⋅m/s)	299.26 ± 63.81	275.61 ± 98.46	291.77 ± 77.14	0.17	0.15
	Ecc:Con Ratio	3 ± 1.98	3 ± 2.51	3 ± 2.18	0.71	0.13
CMJND	Jump Height (m)	0.13 ± 0.06	0.13 ± 0.06	0.14 ± 0.05	0.40	0.08
	mRSI	0.16 ± 0.09	0.14 ± 0.07	0.17 ± 0.08	0.21	0.11
	Force at Min. Displacement (N)	877.68 ± 258.69	832.56 ± 252.16	892.33 ± 250.28	0.10	0.19
	Countermovement Depth (m)	0.15 ± 0.06	0.16 ± 0.06	0.17 ± 0.05	0.29	0.13
	Time to Takeoff (s)	0.91 ± 0.20	0.92 ± 0.16	0.90 ± 0.24	0.88	0.01
	Ecc Time (s)	0.23 ± 0.10	0.25 ± 0.06	0.23 ± 0.09	0.70	0.03
	Peak Ecc Power (W)	303.98 ± 257.85	379.32 ± 186.82	408.33 ± 167.62	0.21	0.16
	Mean Ecc Power (W)	223.08 ± 180.75	259.76 ± 137.85	286.62 ± 122.13	0.23	0.13
	Ecc Impulse (kg⋅m/s)	166.83 ± 63.48	170.25 ± 47.14	168.58 ± 64.71	0.96	0.00
	Con Time (s)	0.32 ± 0.08	0.34 ± 0.06	0.35 ± 0.13	0.48	0.07
	Peak Con Power (W)	1692.37 ± 742.40	1665.54 ± 784.90	1720.82 ± 646.90	0.67	0.02
	Mean Con Power (W)	831.34 ± 418.44	783.17 ± 389.85	850.03 ± 355.66	0.27	0.11
	Con Impulse (kg⋅m/s)	280.43 ± 79.54	281.23 ± 81.42	297.14 ± 100.08	0.47	0.07
	Ecc:Con Ratio	3 ± 2.48	3 ± 2.27	3 ± 2.78	0.43	0.07

Data Represented as Mean ± SD. Bold text signifies significant differences. Abbrevaitions: mRSI= modified reactive strength index. Ecc = Eccentric. Con = Concentric. CV= Coefficient of Variation. ICC = Intraclass Correlation Coefficient. LOA = Limit of Agreement. m = metres. N = Newtons. s = seconds. W = Watt. kg⋅m/s= kilogram meter per second. * = Significant differences between measures IPG and Post. ^ = Significant differences between measures Post and 48Post.

## Data Availability

Not applicable.

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
