# Peer review of "The Effects of Soccer Specific Exercise on Countermovement Jump Performance in Elite Youth Soccer Players"

_children, 2022, doi:10.3390/children9121861_

Round 1

Reviewer 1 Report

Dear authors,

This study is focuses in one of the most studies for its practical application in the assessment of performance, fatigue and training load in athletes, CMJ. Frequently, it´s study only the maximum height reached in the CMJ, but it exists a lot of kinetics and kinematics variables that influence directly in the neuromuscular response. There are some concerns that necessary must be revised:

-  Extension of the introduction section is excessive. Therefore, it must be reduced at least a 50%. It must be avoided the redundant information and to me more concrete.

-    Material and methods section is very confused. It´s necessary to divide this section in different subsections (to use subtitles).

-      It´s necessary to include a Figure that summarize the experimental design of this study. In fact, I propose to divide the experimental section subsections inn study A and study B.

-     Was the experimental sessions at the same time of the day? Based on the circadian influence on CMJ performance, it´s important to clarify it.

-        It´s necessary to specify the type of force platform used.

-   How was it calculated the time of the eccentric and concentric contractions? It has been defined different procedures in the literature, and the type of procedure could influence in the final results.

- The protocol for analysing the response to a soccer protocol must be extended. Currently, it´s not reproducible.

-        Statistical treatment information is very confusing. Authors must rewritten into an only paragraph without redundant information (it´s repeated some procedures).

-        Bland-Altman must be included as graphics. Authors must select the variables based on ICC.

-        In the Table 2 must be reflexed the statistical differences from the Post-Hoc.

-        Development of discussion is very confused. I suggest to discuss variable to variable.

Reviewer 2 Report

Some sentences can be restructured to facilitate the readers' understanding on the content. 

The linkage between soccer and CMJ was only addressed in few sentences of the "introduction". More literature can be reviewed in this area, so that reader can see the significance of this study. 

Round 2

Reviewer 1 Report

Dear authors,

Thank you very much for considering my suggestions. Before an acceptation, you only need to consider the next two minor revisions:

1. To adapt Table 1 to the guidelines of the journal.

2. To increase the quality of the Figure 2.
